# Comparison of Distributed Fiber Optic Sensing and Digital Image Correlation Measurement Techniques for Evaluation of Flexural Behavior of CFRP-Prestressed Concrete Beams

**DOI:** 10.3390/s25237357

**Published:** 2025-12-03

**Authors:** Agnieszka Wiater, Dominika Ziaja, Maciej Kulpa, Tomasz Siwowski

**Affiliations:** 1Department of Road and Bridges, Rzeszow University of Technology, 35-084 Rzeszow, Poland; kulpa@prz.edu.pl (M.K.); siwowski@prz.edu.pl (T.S.); 2Department of Structural Mechanics, Rzeszow University of Technology, 35-084 Rzeszow, Poland; dziaja@prz.edu.pl

**Keywords:** distributed fiber optic sensing, digital image correlation, CFRP-prestressed concrete beam, three-point bending test, strains, displacements, crack development, failure modes, monitoring system

## Abstract

**Highlights:**

**What are the main findings?**

**What are the implications of the main findings?**

**Abstract:**

The paper presents two innovative measurement methods for assessing the flexural performance of precast concrete beams that are prestressed with Carbon Fiber-Reinforced Polymer (CFRP) bars. Strains, displacements, crack development, and failure modes were recorded until failure occurred, using a combination of Distributed Fiber Optic Sensing (DFOS) and Digital Image Correlation (DIC) techniques. This approach provides a more comprehensive understanding of the behavior of CFRP-prestressed beams under load, allowing for more accurate predictions than traditional measurement systems. By integrating both techniques, it is possible to validate results and establish an effective monitoring system. Based on tests conducted on three CFRP-prestressed beams subjected to three-point bending, general recommendations are made for both DFOS and DIC measurement techniques for reinforced concrete (RC) members. DFOS is more effective at detecting minor strains, while DIC excels in measuring large strains in RC structures. Both DFOS and DIC techniques facilitated the monitoring of crack development in RC structures, providing detailed analyses of the location, number, spacing, and width of the cracks. However, beyond the cracking load, the DFOS results can become unreliable due to the impact of cracks on the fiber optic readings. Therefore, DFOS and DIC measurement techniques can be used complementarily, taking into account their respective limitations.

## 1. Introduction

The ongoing development of concrete structures—both reinforced concrete (RC) and prestressed concrete (PC)—and the necessity for their continuous monitoring during operation are key reasons for the search and advancement of measurement methods. These methods enable the easy (remote), accurate, reliable, fast, and cost-effective measurement of fundamental parameters in loaded structures. Key physical parameters that are measured in both laboratory and field conditions include strains, displacements, and the development and morphology of concrete cracking. By assessing these parameters, it becomes possible to evaluate quantities that characterize the behavior of concrete structures under static or dynamic loads. This includes the load–displacement (or strain) relationship, cracking moment, crack morphology, ultimate moment or failure load, and mode, as well as various dynamic characteristics of the structure. Traditionally, these measurements were conducted using foil strain gauges for strains, dial or inductive sensors for displacements, Brinnel magnifiers for cracks, or accelerometers for dynamic characteristics. However, there have been significant advancements in techniques for measuring the performance of concrete structures in recent years. New types of sensors that leverage various mechanical and physical phenomena have emerged. Currently, there are various new measurement methods available, including both destructive and non-destructive (NDT) tools for short-term measurements, as well as methods designed for long-term structural health monitoring (SHM). The development of fiber optic sensing technology and vision sensing technology has further propelled advances in structural health monitoring. Among these innovations, distributed fiber optic sensing (DFOS) and digital image correlation (DIC) techniques stand out as the fastest-growing and most promising methods. 

Distributed fiber optic sensing (DFOS) is a measurement technique that utilizes fiber optic technology [1,2]. This method allows for the measurement of various physical quantities such as strain [3], displacement and shape [4], and temperature [5] along the entire length of optical fibers or sensors [6]. DFOS can effectively replace thousands of traditional strain gauges with a single optical fiber or sensor [7], offering new and innovative possibilities for analyzing and assessing the behavior of civil engineering structures or structural members made from different materials [8,9], including concrete [10,11,12,13], steel [14,15,16,17,18], and FRP composites [19,20,21,22]. While DFOS presents significant opportunities, fully leveraging its advantages requires the use of appropriate sensors or fibers, as well as installation methods that ensure accurate strain transfer from the monitored element to the measuring fiber core [23]. For practical applications in concrete structures, composite monolithic distributed fiber optic sensors are particularly suitable [24,25]. These sensors are produced using a pultrusion process that integrates optical fiber within a composite core. This design protects the optical fiber from damage during operation within the concrete element and eliminates layers that might impede the transfer of deformations, thereby ensuring that the quality of the obtained results is reliable.

Digital Image Correlation (DIC) is a non-contact, vision-based optical measurement technique used to determine the shape, displacement, and strain on the surface of a structure by analyzing digital images. This technique involves comparing a reference image of the surface to subsequent images taken as the structure deforms, enabling the tracking of surface features—often represented by a speckle pattern—to calculate displacements and strains. By comparing the random patterns captured in these images, DIC systems provide detailed information about changes in displacement and strain fields. One of the primary advantages of this measurement method is its capability for full-field observation. Depending on the system setup, both 2D and 3D DIC can be utilized [26]. The number of synchronized cameras observing the same area is crucial for accurate measurements. With only one camera, it is possible to determine the displacements or strains on a single surface. However, using two or more cameras allows for the measurement of out-of-plane displacements. DIC has been applied to a variety of tasks, including both destructive and non-destructive tests, across macro and micro scales. It is particularly useful for understanding cracking mechanisms in concrete [27,28,29], observing changes in crack openings [30,31,32], and measuring concrete strains [33]. Additionally, DIC is employed in the testing of various structural materials such as steel [34], fiber-reinforced polymer (FRP) composites [35,36], aluminum [37], and even asphalt concrete [38]. Its ability to measure displacements in all three dimensions has also been applied in inventory management, field testing, and structural health monitoring (SHM) of bridges [39,40,41,42], tunnels [43,44,45], and other significant engineering structures [46].

This paper covers part of the research conducted by Rzeszów University of Technology (RUT) on the development and implementation of a novel method for prestressing precast concrete beams using carbon fiber-reinforced polymer (CFRP) bars. CFRP-prestressed concrete structures have lower deformation capacity than CFRP-reinforced concrete structures. To address this, high-strength concrete (HSC) was used to enhance the flexural performance of the prestressed beams. The research aimed to assess the flexural behavior of precast prestressed concrete beams using a new CFRP prestressing technique. In this preliminary study, three high-strength concrete CFRP-prestressed beams were tested for flexural performance, with loading, strains, displacements, cracks, and failure modes recorded using three different measurement methods. The DFOS and DIC techniques monitored strains, cracks, and failure modes under load and verified the accuracy of the indirect deflection measurements obtained with these methods. Traditional linear variable differential transformers (LVDT) sensors were also utilized.

The scientific focus of this paper is on combining DFOS and DIC to record strain and crack data in RC structures. According to the literature review (see p. 2), using both DFOS and DIC provides a more comprehensive understanding of how loaded structures behave. This, in turn, allows for more precise predictions of performance that traditional measurement systems cannot achieve. Furthermore, integrating both techniques enables result validation and the development of an effective monitoring system. The aim of this research is to assess the synergy of DFOS and DIC on a typical prefabricated RC beam, which features innovative materials and structural solutions (CFRP, HSC, prestressing) that are not relevant to this study. The main objectives are: to verify the measurement accuracy of both methods when used together; to explore the complementarity of results from each system; and to evaluate the applicability of each for assessing RC structure behavior (cracks, strains, displacements). Since the combined use of DFOS and DIC is the primary focus, a detailed discussion on the behavior of the new precast CFRP-prestressed beams under load until failure is provided elsewhere.

## 2. The Combined Application of DOFS and DIC—A Short Review

There have been several experimental studies examining the combined use of DOFS and DIC for strain measurement in RC structures. For instance, Bado et al. [47] presented the results of an experimental test involving two RC tensile elements equipped with DOFS-instrumented steel rebars, which provided internal strain readings. These internal measurements were integrated with external monitoring from DIC, which tracked the displacements and strains on the surfaces of the elements. The authors demonstrated that using both DOFS and DIC technologies together resulted in an excellent monitoring system. This system allowed for continuous interaction and verification between the internal and external strain readings, offering a comprehensive view of the strain distribution within the elements. Similarly, Mata-Falcón et al. [48] investigated the use of combined DFOS measurements on steel reinforcement and DIC measurements on concrete surfaces. They highlighted several advantages of this integrated approach: (i) mutual validation of results, (ii) linking the stress distribution in the reinforcement to the kinematics of the concrete, and (iii) analyzing how load distribution varies in different parts of the element over time. A significant contribution of their work was the discussion of measurement uncertainty assessment for DIC and DFOS in large-scale structural tests. The experimental results revealed a strong correlation between the average fiber strain measurements and the DIC results, with crack locations predicted by both measurement methods aligning perfectly.

The measurement methods for recording strain considered in the work [49] represent both well-established techniques (strain gauges) and novel techniques (DFOS and DIC). Two test series were conducted, one on RC tension rods and another on an RC beam subjected to a four-point bending test. From the latter scenario, certain generalizations were to be deduced for varying load levels: low strains are measured accurately using the DFOS technique. Conversely, DIC was found to be an adequate choice for assessing higher strain levels and concomitant concrete cracking, as this non-contact technique avoids the imprecision caused by adhesives. A recent testing campaign conducted by Hoult et al. [50] used distributed fiber optic sensors embedded along several longitudinal steel rebars of the RC U-shaped walls. The resulting experimental dataset provides an opportunity to evaluate and compare the strain measurements obtained with the DFOS technique against more conventional and state-of-the-practice sensors that monitor surface displacements and deformations. This work highlighted the need to average strain measurements from DIC techniques to obtain coherent results with those measured from fiber optics.

Both techniques were frequently tested together for crack formation and monitoring, as well as for SHM of concrete structures. For instance, Herbers et al. [51] conducted experimental investigations to assess the suitability of five different types of DFOS for crack monitoring. The DFOS measurements were validated using DIC and electrical strain gauges. The various DFOS types exhibited noticeable differences in the measured strain curves and the determined crack widths. In comparison to the layered sensing cable, a monolithic DFOS provided a more precise strain distribution, displaying pronounced strain peaks even for closely spaced cracks. The crack widths obtained by integrating the strain curves showed a strong correlation with those measured through DIC.

Additionally, the study [52] employed different NDT methods to analyze the load-bearing behavior of an RC beam under a four-point bending test. The focus was on determining failure modes using various optical NDT techniques, including DFOS and Fiber Bragg Grating (FBG). Furthermore, optical deformation measurements were conducted using DIC and stereophotogrammetry (SP). The results from the laboratory tests on the RC beam demonstrated that optical measuring systems are effective tools for SHM. The three-dimensional detection of deformations and resulting strains through the DIC technique is advantageous for analyzing the damage mechanisms of loaded structures and related crack formation. The same applies to strain measurements using DFOS. The authors concluded that the combination of DIC and DFOS systems offers new opportunities for SHM that are not achievable with traditional measuring systems.

The last paper mentioned here [53] summarizes all achievements to date concerning two sensing technologies: fiber optic sensing technology (both discrete and distributed) and vision sensing technology (DIC). This study conducts a comprehensive review of the literature, focusing on the fundamental principles, recent advancements, and current applications within this field. It outlines the advantages and limitations of both sensing technologies and discusses potential future directions. Furthermore, the paper explores the integration of fiber optic sensing technology with vision sensing technology, providing numerous examples to illustrate this approach. The conclusions indicate that this new integrated sensing technology can effectively leverage the strengths of both fields.

In summary, the main feature of DFOS technology is its ability to perform contact measurements, while DIC technology utilizes non-contact measurements. Both types of measurements—contact and non-contact—are essential for a more accurate evaluation of a structure’s health. By integrating these two techniques, we can cross-verify and validate the results obtained from each method. Furthermore, this combination allows for the integration of external strains measured through vision sensing techniques with internal strains measured by fiber optic sensing techniques. This comprehensive approach provides more detailed health information about the structure and leads to more accurate predictions of its performance under load.

## 3. Materials

High-strength concrete C60/75 was employed for the precast beams. The composition of the concrete is listed in Table 1. In designing the beams, the following material parameters were utilized according to Eurocode 2 [54]:concrete class: C60/75:
-compressive strength f_cd_ = f’_c_ = 60.0 MPa,-modulus of elasticity: E_cm_ = 36.4 GPa,-ultimate compressive strain: ε_cu_ = 0.003,exposure class: XC4, XF4;chloride content class: Cl 0.20;cement type: CEM II/A-M (S-LL) 52.5N;aggregate: dolomite; maximum aggregate size D_max_ = 16 mm;W_eff_/C ratio: 0.38;consistency class: S4.

The CFRP composite bars designed for prestressing, featuring a profiled round shape with spiral ribbing, were supplied by the manufacturer. Since no material testing was conducted on the CFRP bars, the following minimum material characteristics were assumed for the design of the beams, based on the manufacturer’s specifications:fiber volume fraction in CFRP bars: ≥60%;characteristic short-time tensile strength for the nominal cross-sectional area: ≥2100 MPa;average short-time tensile strength for the nominal cross-sectional area: ≥2500 MPa;average modulus of elasticity for the nominal cross-sectional area: ≥155 GPa.

The remaining reinforcement for the beams utilized glass fiber-reinforced polymer (GFRP) bars. These GFRP bars were manufactured with a spiral wrap, which enhances the bond performance with the concrete matrix. According to the manufacturer’s specifications, the reinforcement has a guaranteed tensile strength of 1100 MPa.

Since there are no specific standards for designing concrete beams prestressed with CFRP bars, relevant provisions from various established standards were utilized for the design of the test beams. The primary basis for the design was the American standard ACI 440.4R-04 [55], while additional input was drawn from several other standards, including ACI 440.1R-15 [56], ACI 318-19 [57], CAN/CSA S6-19 [58], CAN/CSA S806-12 [59], and Eurocode 2 [54].

For the design, a rectangular cross-section was chosen, considering the following recommendations from the codes:according to Eurocode 2 [54], the recommended beam depth (H) for supported beams should fall within the range of H ≥ l/20 L and H ≤ l/14 L;according to ACI 318-19 [57], the recommended beam depth (H) should be H = l/16 L;the recommended ratio of depth to width (H/B) according to ACI 440.1R-15 [56] should be maintained within the range of 1.5 to 2.

Three high-strength concrete beams, prestressed with CFRP bars, were prepared for testing (see Figure 1). The length of the beams was chosen to ensure compliance with engineering practices and laboratory standards, while also accurately representing the behavior of real structures in a laboratory setting. This length allows for a high bending moment paired with relatively low shear forces. Ultimately, the beams were designed with a length of L = 480 cm, a depth of H = 27 cm, and a width of B = 15 cm, meeting the necessary recommendations. The effective length of each beam, Lt, is 450 cm, accounting for a 15 cm overhang at both supports.

Five CFRP bars, each with a diameter of 6 mm, were utilized as the main beams’ prestressing for flexural strength, arranged in two rows. Due to technical constraints, a maximum of three bars could be placed in a single row. The vertical distance between the two rows was set at 43 mm, dictated by limitations associated with the new prestressing device. The system was designed to prestress the bars with a force equivalent to 60% of the characteristic short-term tensile strength of CFRP. This value represents the maximum allowable jacking stress for CFRP, as specified in ACI 440.4R-04, Table 3.3 [55].

The prestressing force was established at 35.6 kN per bar, resulting in a total initial force of 178.1 kN per beam. Design calculations indicated that the total prestressing losses would amount to 18.78%, yielding a final prestressing force of 144.7 kN per beam.

For the top reinforcement of the beams, two GFRP bars with a diameter of 12 mm were used along the entire length of the beam. Additionally, two-legged GFRP stirrups with a diameter of 4 mm were applied at 5 cm intervals to minimize shear effects on bending. To accommodate thermal expansion and the Hoyer effect, both the top and bottom concrete cover in the beams was set to 40 mm, in accordance with CAN/CSA S806-12 [59].

Three concrete beams were manufactured without permanent anchorages at the ends, similar to those used in post-tensioned structures. Instead, prestressing relied on the bond between the CFRP bars and the concrete. Given the brittle nature of CFRP bars, a specialized prestressing procedure was necessary. Custom-designed clamps were used to securely fasten the CFRP bars without causing damage or crushing them.

A specially designed self-supporting formwork was created to cast the beams, allowing the prestressing process to proceed smoothly while withstanding the prestressing forces. Each bar was tensioned individually, and the prestressing force was continuously monitored using a strain gauge, with readings taken every minute.

One day before casting the beams, the CFRP bars were prestressed with an initial force of 30 kN per bar to prevent slippage from the clamps and to identify any defects in the bars. Immediately after casting, the CFRP bars were tensioned to the required design prestressing force, and the concrete was thoroughly compacted to ensure proper integrity and optimal bonding with the bars.

## 4. Methods

The beams were tested using a three-point bending scheme, supported by two steel rollers that were spaced 4.5 m apart (see Figure 2). A hydraulic actuator with a maximum loading capacity of 630 kN was used to apply the load. This load was applied at a constant rate of either 1.0 or 0.5 mm per minute, utilizing displacement control. The beams were subjected to static loading in seven sequential steps until failure occurred. The load levels were determined based on the design calculations outlined in Table 2.

The beams were designed so that their bending capacity is primarily determined by the tensile strength of the bars, classifying the section as tension-controlled. In this scenario, the strain in the concrete should not exceed 0.003 at the moment of the beam’s failure. The nominal bending capacity listed in Table 2 was calculated based on the flexural strength according to ACI 318-19 [57] and was not reduced by the strength reduction factor (ϕ). The ultimate tensile strength of CFRP tendons (f_pu_) represents the average short-term tensile strength of the material. The design bending capacity in Table 1 is a factored value derived from the nominal flexural strength, which is multiplied by the strength reduction factor (ϕ).

Three measurement methods were employed to monitor the behavior of the beams under subsequent test loads: DFOS for measuring concrete strain, monitoring the initiation of concrete cracking, and assessing beam displacement; DIC for tracking concrete strain, observing the development of concrete cracking, and measuring beam displacement; and LVDT sensors for calibrating DIC and measuring beam displacement. The locations of all sensors are illustrated in Figure 3.

The standard telecom fiber, covered with a soft acrylic coating and capable of being tensioned to over 5% [19], was used for the DFOS measurement. The fiber optic sensors were attached using epoxy adhesive on both sides of each beam at two locations along its entire length: at the top and bottom reinforcement levels. Concrete strains were measured at intervals of 2.6 mm, resulting in more than 1800 measurement points along the beam’s length. For DFOS measurement, the optical backscatter reflectometer ODiSI-B by Luna Inc., (Roanoke, VA, USA) was utilized, operating at a frequency of 0.5 Hz. The key measuring parameters relevant for data analysis are summarized in Table 3.

The objective of the DFOS measurement was to obtain the distribution of concrete strains along the entire length of the beam under a controlled load. Strain measurements in the tension zone of the beam were conducted accurately until cracking initiated, allowing for the identification and monitoring of crack development. However, as the load increased and cracks formed, the concrete strain values in the cracked sections became unreliable. In contrast, strain measurements in the compression zone of the beam remained valid until the beam failed.

Moreover, since strain measurements were taken simultaneously at two parallel levels (the top and bottom of the beam), it was possible to calculate the rotation angles of individual cross-sections of the beam and subsequently determine the vertical displacements within each cross-section. The method for calculating vertical displacements from the DFOS strain measurements is detailed in the paper [19].

The DIC measurement was conducted simultaneously with DFOS recording. The DIC results provided a full-field measurement of strains and displacements by analyzing digital images of a deforming beam. This non-contact optical technique allows for a detailed analysis of how the beam’s surface changes over time or in response to different loading steps. Unlike traditional point-based measurement methods, DIC collected data across the entire selected surface of the beam, offering a comprehensive view of deformation. The measurements focused solely on one side of the beam, specifically in the area from 1.6 m to 2.8 m along its length. The grid of points used to obtain displacements and strains through DIC measurements is illustrated in Figure 4.

The Q400 system from Dantec Dynamics GmbH (Ulm, Germany), combined with Istra4D V4.10 x64 software and two Baumer 12.3 Mpx cameras equipped with VS-1620HV (VS TECHNOLOGY CORPORATION; Tokyo, Japan) lenses, was utilized for this vision sensing technology. The distance between the tripod and the beam surface was 148 cm, and a Pl-35-WMB_9x9 (delivered as a component of Q400) calibration target was employed. Exemplary views captured simultaneously by both cameras are presented in Figure 5.

During the loading of the beam, a direct connection between the DIC system and the loading machine was not established. As a result, the calibration of DIC measurements with the data recorded from the LVDT was required. To achieve this, three virtual sensor points (1, 2, and 7) were used to link DIC measurements to specific loading steps, as shown in Figure 6. The average vertical displacement measured at these points—two by the LVDT and one by the actuator piston—was found to correspond to designated loading forces. For each test load, as detailed in Table 1, images of the strains on the concrete surface of the beam were captured. The shifts in the positions of points were calculated using DIC algorithms by comparing the reference image (the first image taken in each series) with the image corresponding to the selected loading step.

To monitor DFOS and DIC indirect displacement measurements and to calibrate DIC strain measurements with specific load levels, a set of LVDT transducers was used. Their arrangement is illustrated in Figure 2 and Figure 3. Twelve LVDT transducers were attached to the bottom of each beam to measure vertical displacements (deflections) at four key sections: at mid-span (specifically 15 cm from the load axis towards each support), as well as at the quarter-span and three-quarter-span positions. In these four sections, two transducers were placed on each side of the beams. Additionally, two sets of LVDT transducers were positioned at the support axis to monitor any potential movement of the beam’s supports. To avoid damaging the LVDT transducers during the beam’s failure, they were generally removed at a load level of 22.7 kN (design capacity). To complement the LVDT readings of the load–displacement relationship, the displacement of the actuator piston was also taken into account.

## 5. Results

### 5.1. DFOS Measurement

The DFOS was the main method used to measure concrete strains in the current tests. Figure 7 and Figure 8 illustrate the strain distribution in the example beam across all applied load steps. The results obtained from both optical fibers (upper and lower) on each side of the beam are presented. The strain distribution patterns for the other two beams were very similar. The graphs in Figure 7 and Figure 8 display the concrete strain distribution in both the top and bottom zones of the beam. However, reliable readings could not be obtained in the tension zone for significantly cracked concrete at the 52.4 kN and 65 kN load levels. As noted earlier, the strain values in the cracked sections (specifically, the bottom zone using the lower optical fiber) are unreliable; nevertheless, they can still be analyzed for insights into the development of cracking in the beam. In contrast, the compression (top) zone evidences local stress peaks, which likely indicate that shrinkage cracks are closing as a result of the applied load.

Despite having the same load and identical locations for the optical fibers, the measured values—particularly the strain distributions—differ between the two sides of the beam. These differences are especially noticeable in the lower zone of the beam, where strains exceed the cracking load level. However, this discrepancy is understandable, as strains measured above the cracking load are not deemed reliable. Before cracking occurs, the strains on both sides of the beam are similar, which may suggest the applicability range of the DFOS measurement technique in concrete structures. This observation is supported by the strain readings from both sides in the upper zone, where they are nearly identical, with the exception of two or three sections. The nature of these strain peaks, which correspond to the closure of shrinkage cracks, helps clarify the observed differences. This is because the distribution of shrinkage cracks can vary randomly on either side of the beam.

The indirect method for determining the vertical displacements (deflections) of beam sections, based on DFOS strain measurements, was employed. This approach allowed for the determination of the complete deflection profile of the beams under various load steps. Since the optical fibers were positioned on both sides of each beam, deflection plots were generated for both sides, and the final plot was created by averaging the results. Figure 9 illustrates the deflection lines of one of the beams based on DFOS measurements taken from both sides. The deflection lines derived from these strain measurements are consistent in both shape and magnitude, confirming the high reliability of the algorithm used to calculate beam deflections based on DFOS technology [19].

### 5.2. DIC Measurement

Key measurement results from DIC were obtained in the form of displacement and strain maps (see Table 4). For displacement measurement, DIC quantified the movement of points on the surface of a beam between different images. By analyzing the displacement field, DIC was then able to calculate strains, which represent the deformation of the beam’s material (concrete).

An example diagram of the concrete strain distribution for a selected beam in the bottom zone (corresponding to sensor line 1, see Figure 6) is presented in Figure 10. Strains were recorded only on the portion of the beam’s surface covered by the DIC measurement, specifically between 1.6 m and 2.8 m along the beam’s length. The highest load level before failure, which was 52.4 kN, was selected for this presentation. This choice is significant because, at high loads—when concrete strains are substantial—the measurement errors associated with DIC (which tend to increase with larger observed areas) are relatively small. As a result, the strain values obtained can be regarded as reliable. The concrete strains in the top zone of the beam have been excluded due to a relatively large measurement error in those values. This error arises from the special positioning of the cameras, which were set up to record the beam’s deflection over the maximum feasible distance, while also considering the limitations of the available equipment and the requirements of DIC measurements.

The diagram of displacements for a sample beam across the DIC measurement surface is illustrated in Figure 11. This diagram displays the deflection lines of the beam for each load step, which were determined using images of two reference points (points 1 and 2, as shown in Figure 6). These reference points are positioned at the level of the DFOS bottom optical fiber. Due to the constraints of the DIC surface area, the deflection line of the beam is only tracked within the cross-section range of 1.6 m to 2.8 m, covering a length of 120 cm. However, since this segment represents the central portion of the supported beam, the most significant deflections occur in this area. This focus allows for easier comparison of the deflections with those measured using other methods.

The accuracy of DIC measurement is primarily determined by the configuration of the measuring station. Key factors include the distance between the cameras and the object being measured, the resolution of the cameras, and the size of the facets (the matrix of pixels in the image that defines each measurement point). In the measurements conducted, the facet size remained constant across all specimens, time steps, and measurement points. In the Dantec Dynamics Q400 system, each measured quantity at every point and time step has a separately calculated measurement error. This error accounts for factors such as the camera’s position, lens distortion, and other variables. It is not appropriate to assign a single measurement error value for all collected data, even if you are analyzing only one quantity. Additionally, the need for stable lighting conditions is crucial in the application of DIC measurement. Variations in lighting, such as those caused by a sunny day, can lead to overexposure of the images, resulting in the loss of data.

### 5.3. LVDT Deflection Measurement

The direct measurement of displacement using LVDT was conducted to validate the measurements obtained from the DFOS and DIC techniques. Figure 12 presents the deflection measurements of the selected beam taken at four different sections. The plot includes data from five different load levels. LVDT measurements indicated that the beams displayed remarkably uniform behavior under load in a three-point bending setup (see Figure 12). The deflections of the beams on either side of the load axis were nearly identical, demonstrating the good uniformity of the concrete, the effectiveness of the prestressing force, and the high quality of the beam casting.

### 5.4. Failure Mode

The beams failed at load levels of 69.67 kN, 68.82 kN, and 73.14 kN for beams 1, 2, and 3, respectively. The average ultimate load was 70.54 kN, which is 35% higher than the theoretical nominal carrying capacity of the beams. This discrepancy may be attributed to two main reasons: the actual compression strength of the concrete in the beams was greater than the assumed concrete class in the calculations, and/or the tensile strength of the CFRP bars exceeded the nominal values declared by the manufacturer. The latter reason appears to be more plausible. The typical failure mode observed in CFRP-prestressed beams was a sudden onset of cracking that progressed until the CFRP bars broke. This was followed by concrete crushing in the beam’s top (compressive) zone and an increase in the beam’s deflection beyond the critical value (see Figure 13).

## 6. The Comparison of DFOS vs. DIC Measurement

### 6.1. Strains

In Figure 14, the comparison of strains recorded using two methods on the bottom line of the beam is presented (refer to Figure 6). As discussed previously, the DIC measurement was initially set up to determine displacement, but it lacks sufficient accuracy for strain measurement at such low levels of load. The errors are larger at the edges of the observed areas compared to the center. Between the cracks, compressive stresses occur “incidentally,” and their values fall within the measurement accuracy limits. For DFOS, which was primarily applied for strain measurement, a limitation is its response to high strain measurements. Due to these two factors, two separate scales were used: the left side for DFOS (under loads between 11.6 kN and 22.7 kN) and the right side for DIC measurements (under loads of 11.6 kN and 52.4 kN). The DIC measurement for 11.6 kN shows values that are consistent with measurement accuracy. At higher load levels, the sections where cracks appeared were clearly indicated (the peaks are visible).

It is important to emphasize that the recorded strains correspond to the strains in the concrete, particularly in relation to the cracks. This indicates that the fiber in the DFOS adheres to the beam along its entire length, with no discontinuity patterns caused by the crack. After the appearance of cracks, the strains obtained via DIC in the vicinity of the cracks can be used to estimate their width. However, it is impossible to achieve a concrete elongation of 15%, as observed in the loading case of 52.4 kN for the beam cross-section with a length of 2.1 m. This result is due to DIC algorithms, which, up to a certain point, recognize the similarity of pattern parts representing one point on both sides of the crack, thereby artificially increasing the reported strain or elongation.

Both measurement techniques produce qualitatively comparable results regarding the location of the cracks. The local peaks are found in similar locations when comparing the DFOS measurement at 22.7 kN and the DIC measurement at 52.4 kN. These techniques can be used complementarily, taking into account their respective limitations: DFOS is more suitable for detecting minor tensile or compressive strains, while DIC is better for larger strains.

For a more detailed analysis of the strain changes during the test, the strains measured at two points are presented in Figure 15. These points were chosen at the upper and lower DFOS located at the mid-span of the beam. This allows us to analyze the beam’s response to the applied load. It can be observed that up to a load of 30 kN, the beam behaved elastically and exhibited linear characteristics. Beyond this point, a noticeable decrease in the beam’s stiffness was recorded. In the compressed section, the stresses increased in a nearly linear manner, although there were notable anomalies, including the transition from cracks to fractures. Before failure occurred, there was a non-linear increase in strain observed on one side of the beam, indicating an imminent failure. In the tensile zone, following the initial cracking, strain increased irregularly. This irregularity resulted from cracks developing in a somewhat random and uneven manner on both sides of the beam. The flat sections on the graph signify a sudden increase in strain, which is a result of the brittle fracture of the concrete.

Summarizing the strain measurement, it should be noted that the measurement accuracy of DFOS and DIC at low strain levels needs validation against objective third-party data (e.g., local strain gauges), rather than relying solely on data stability. Additionally, the processing method for raw DIC data can affect the representation of minor strains; therefore, claiming that DFOS is more accurate than DIC at low strains is not rigorous.

### 6.2. Displacements

A qualitative comparison of the deflection of a tested beam was conducted using three different methods: the direct method (LVDT) and two indirect methods (DFOS and DIC). The results are illustrated in Figure 16. Reliable measurements from the DFOS and LVDT methods were only obtained for loads up to 22.7 kN; therefore, the comparison was limited to loads below this threshold. The deflection distributions along the beam, as determined by the three methods, were consistent with one another. However, as the load increased, the discrepancies among the methods also became more pronounced.

At the highest load levels, deflection measurements were exclusively possible with the DIC method, as shown in Figure 17. For these higher loads, the DFOS measurements became unreliable due to the emergence of cracks in the beam. Additionally, the LVDT sensors had to be removed to prevent potential damage in the event of a sudden beam failure.

The load–displacement relationship for the first beam tested is presented in Figure 18. Since this beam was the first examined and its destruction mechanism was unknown, the testing equipment was relocated before reaching the final stage to avoid damage. The results obtained were consistent with the readings from the machine’s piston displacement sensor. The displacement resulting from the increase in load is illustrated in Figure 18. The graph displays the typical behavior of RC structures. Notably, after the first crack occurs at approximately 30 kN, the stiffness of the beam decreases. This reduction in stiffness leads to further damage, which is represented as spikes in the plot beyond that point.

The comparison of results from three different methods is presented in Table 5. To enhance the evaluation of deflections at the highest load level, we also included the deflections measured by the actuator piston of the loading machine. This measurement is taken directly on the beam’s top surface, differing from the measurements obtained using LVDT sensors. The variance between this direct measurement and the DIC measurement is significant, although it diminishes as the load increases. The DIC measurement was calibrated against the displacements recorded by the LVDT sensors, resulting in excellent compliance. This indicates that a well-calibrated DIC measurement can reliably substitute for direct measurement at higher load levels, particularly when LVDT sensors are removed. In the case of DFOS measurements, satisfactory compliance was achieved, but only up to the beam’s cracking load of 22.7 kN. Beyond this cracking load, the DFOS results become unreliable due to the influence of cracks on fiber optic readings.

### 6.3. Crack Development

Both DFOS and DIC measurement techniques have a significant advantage: they enable the monitoring of crack development in structures. Based on the data obtained from these measurements, a detailed analysis can be conducted regarding the number, spacing, and width of the cracks. In DFOS, diagrams generated for the bottom (tension) zone illustrate the locations of the cracks and the load levels at which the beams began to crack, as shown in Figure 7 and Figure 8. These diagrams provide a quantitative depiction of crack development. The DFOS technique measures strains along the entire length of the beam with relatively frequent sampling (e.g., 0.5 Hz), allowing for an in-depth analysis of crack formation and propagation. Strain measurements can be conducted until the crack crossing the fiber optic sensor reaches a width of approximately 0.5 mm. Beyond this width, DFOS measurements at the crack location become unreliable, but this limitation is generally sufficient for most practical structural measurements.

The verification of crack locations and their subsequent development can be conducted using DIC measurements. Figure 19 presents a comparison of the crack patterns observed in a selected beam using DFOS and DIC measurements. From this comparison, it is evident that the number of cracks identified by both methods is converging; specifically, the cracks depicted in the DIC image correspond to peaks in the same regions on the DFOS plot, indicating a disturbance in the measurements from the fiber optic sensor.

### 6.4. Failure Mode

The DIC measurement is the only one among the three measurement technologies that enables a more accurate analysis of the failure modes of beams after they have been subjected to ultimate (failure) loads. DIC measurements were conducted until the end of the tests for the second and third beams; for the first specimen, measurements were taken up to the ultimate loading step. The images captured of the beam under ultimate load facilitate an in-depth analysis of its behavior in this critical phase.

Figure 20 presents the effective strain distribution map, which calculates effective strain at each point according to the von Mises formula, alongside a conventional image of the failure mode. Beam No. 3 was selected for this analysis, as it experienced the highest ultimate load of 73.14 kN, allowing for extended DIC measurement time. The failure modes observed in both images were remarkably similar, with consistent numbers, locations, and lengths of vertical cracks. Additionally, horizontal cracks along the bottom reinforcement (CFRP bars) were evident, confirming that CFRP rupture was the primary cause of the beam’s failure.

## 7. Summary and Conclusions

The paper presents two novel measurement methods used to evaluate the flexural performance of precast concrete beams that are prestressed with CFRP bars. In the preliminary study, three identical high-strength concrete CFRP-prestressed beams were subjected to a three-point bending test under displacement control. Measurements of strains, displacements, cracks, and failure modes were recorded until failure occurred, utilizing a combination of DFOS and DIC techniques. This approach provides a more comprehensive understanding of the behavior of loaded CFRP-prestressed beams, allowing for more accurate predictions than traditional measurement systems can achieve. Moreover, by integrating both techniques, one can validate the results and establish an effective monitoring system.

Based on tests on three CFRP-prestressed beams subjected to three-point bending, general recommendations are derived concerning both DFOS and DIC measurement techniques for RC members.

DFOS is more suitable for detecting minor tensile or compressive strains in RC structures.Beyond the cracking load, the DFOS results become unreliable due to the influence of cracks on the fiber optic readings.DIC is a better technique for measuring large tensile or compressive strains in RC structures.At the highest load levels, deflection measurement is exclusively possible with the well-calibrated DIC method, particularly when LVDT sensors have to be removed.Both DFOS and DIC measurement techniques have a significant advantage: they enable the monitoring of crack development in concrete structures. Based on the data obtained from these measurements, a detailed analysis can be conducted regarding the location, number, spacing, and width of the cracks.The DIC measurement is the only one that enables a more accurate analysis of the failure modes of RC beams after they have been subjected to ultimate (failure) loads.

The results showed that both DFOS and DIC measurement techniques can be used together to assess RC structures under flexural loading. However, their respective limitations should be considered. Nonetheless, the specific findings of this research are difficult to generalize quantitatively, as they relate only to the particular case study described. Therefore, the conclusions provide a broad comment rather than a detailed quantitative assessment. Nonetheless, the overall trend is accurate, and we believe it can be presented in this way. The authors also successfully conducted studies on a real bridge structure using both methods. The results of this research, demonstrating the feasibility of employing both approaches to monitor a real bridge, will be published soon.

## Figures and Tables

**Figure 1 sensors-25-07357-f001:**
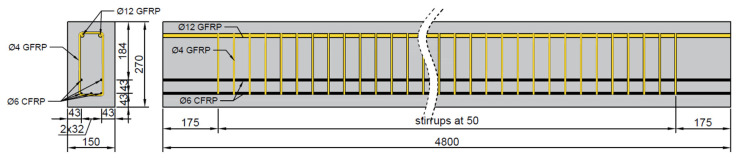
Geometry and reinforcement of the tested beams.

**Figure 2 sensors-25-07357-f002:**
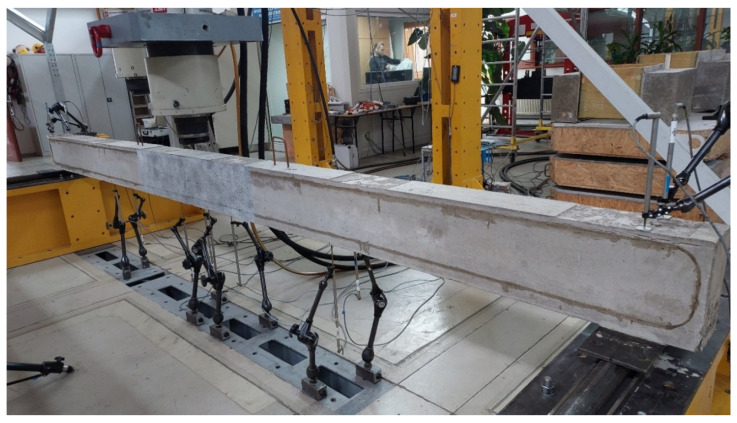
The three-point-bending test setup.

**Figure 3 sensors-25-07357-f003:**
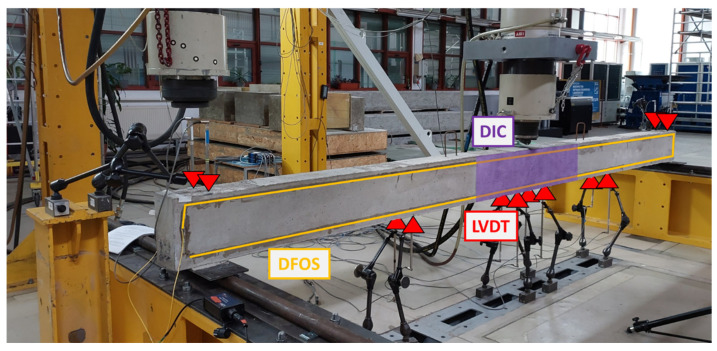
Location of all sensors for the beam testing.

**Figure 4 sensors-25-07357-f004:**
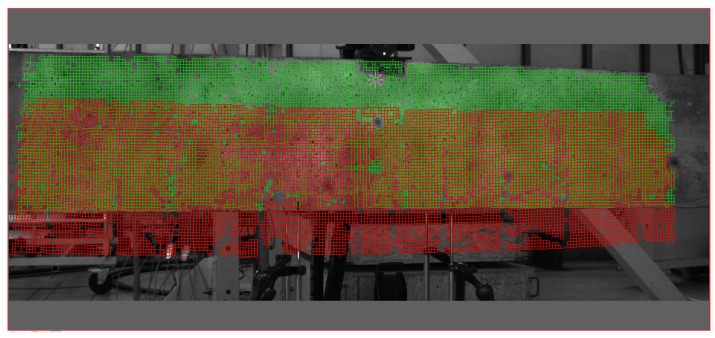
The grid of points for which the displacements and strains were obtained using DIC measurements (green—reference step, red—the final measurement).

**Figure 5 sensors-25-07357-f005:**
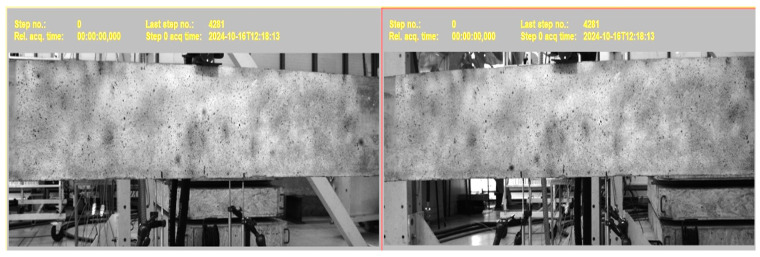
Exemplary views recorded by the cameras during DIC measurements (one selected time step).

**Figure 6 sensors-25-07357-f006:**
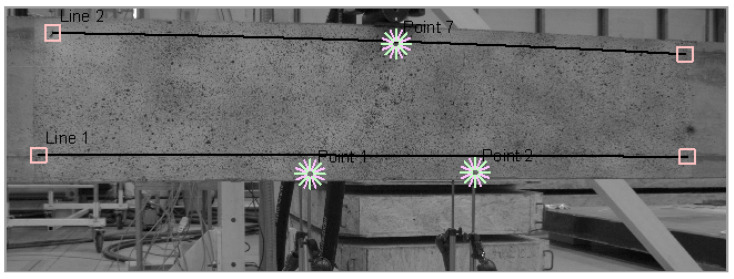
Virtual sensors: points (1, 2, 7) used for calibration of DIC measurements, and line sensors (Line 1, Line 2) used for strain comparison with DFOS.

**Figure 7 sensors-25-07357-f007:**
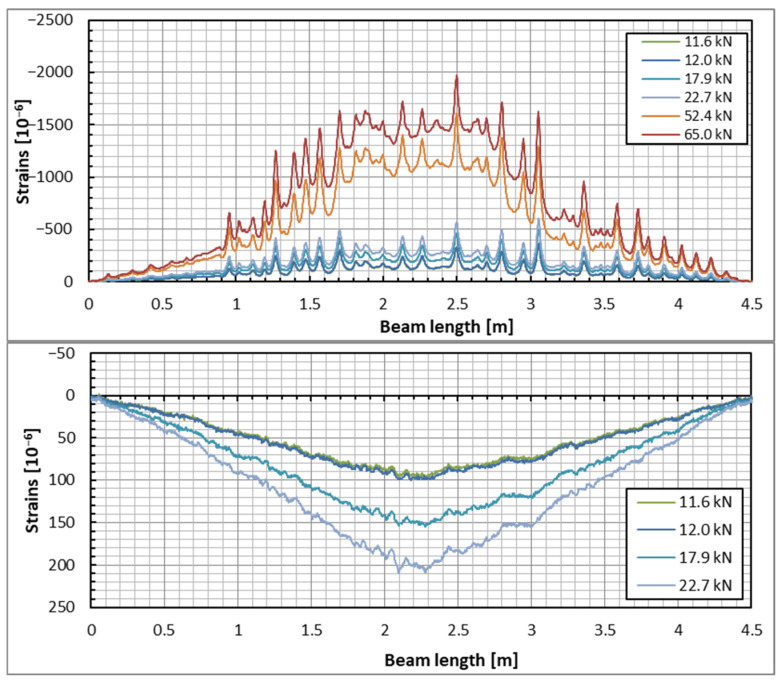
Concrete strain distribution in the top (**upper plot**) and bottom (**lower plot**) zones of the beam under the applied load steps—the beam’s side with the DIC measurement surface.

**Figure 8 sensors-25-07357-f008:**
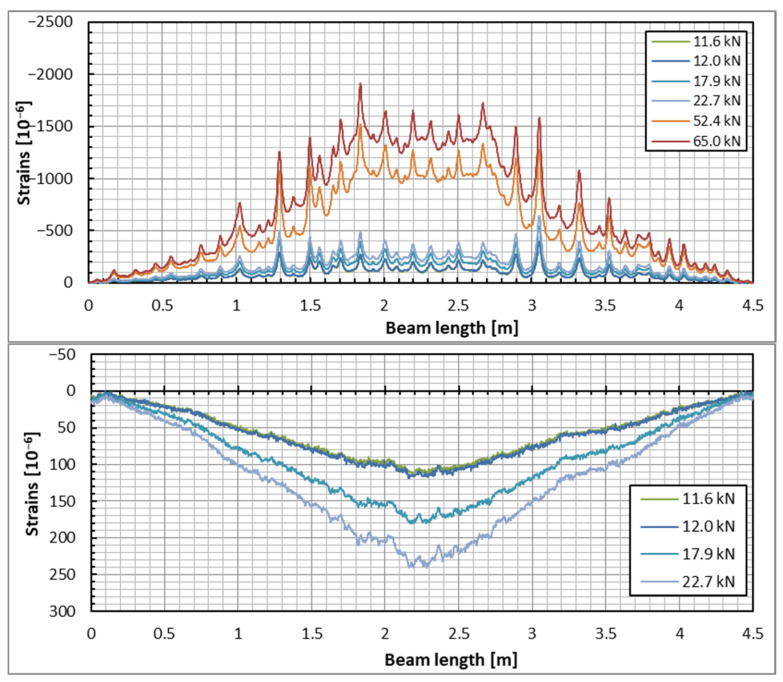
Concrete strain distribution in the top (**upper plot**) and bottom (**lower plot**) zones of the beam under the applied load steps—the beam’s opposite side.

**Figure 9 sensors-25-07357-f009:**
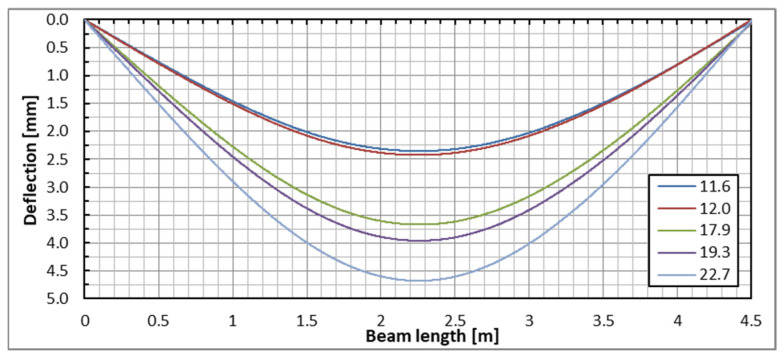
Beam deflection lines under the applied load levels based on the DFOS strain measurement on the beam’s side with the DIC measurement surface (**top**) and on the other surface (**bottom**).

**Figure 10 sensors-25-07357-f010:**
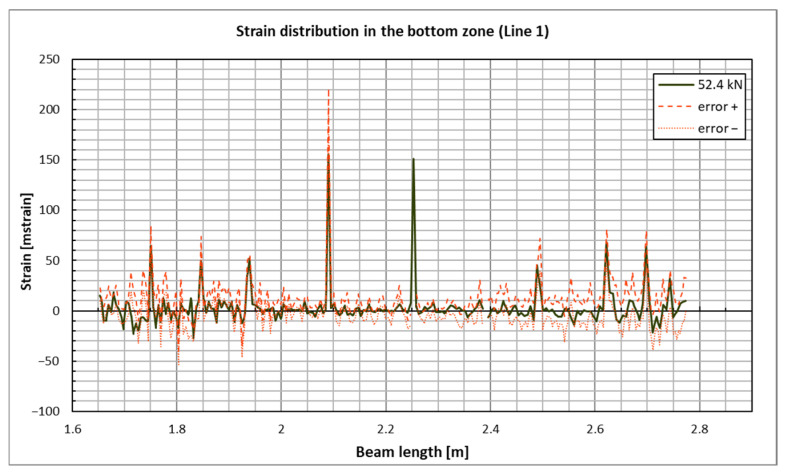
Concrete strain distribution in the bottom zone of the beam under the highest load level—the beam’s side with the DIC measurement surface.

**Figure 11 sensors-25-07357-f011:**
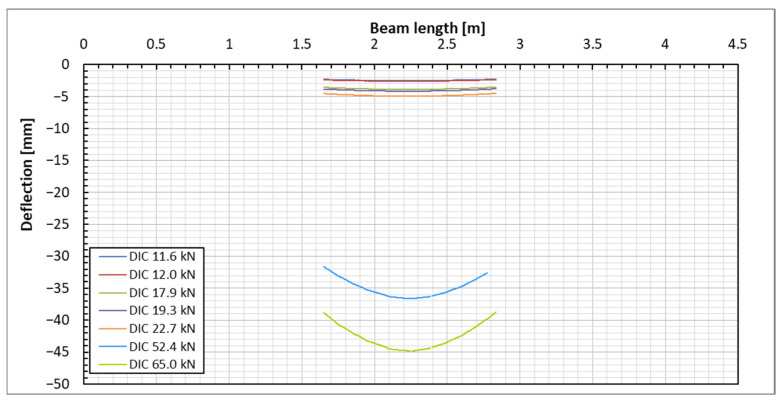
Beam deflection lines under the applied load levels based on the DIC measurement on the beam’s side surface.

**Figure 12 sensors-25-07357-f012:**
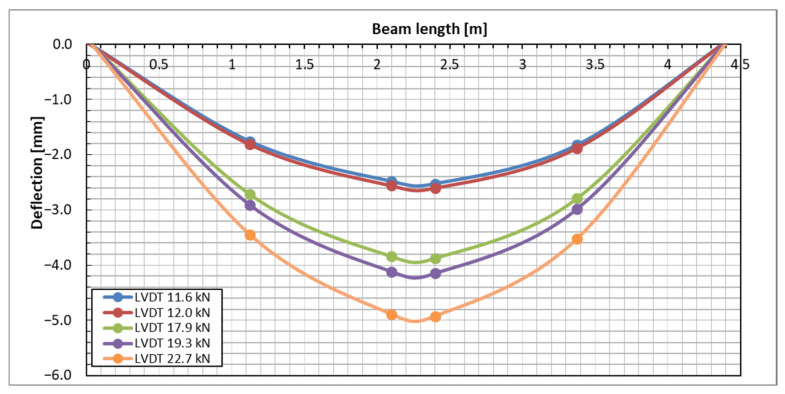
The deflection measured by LVDT sensors under subsequent load levels.

**Figure 13 sensors-25-07357-f013:**
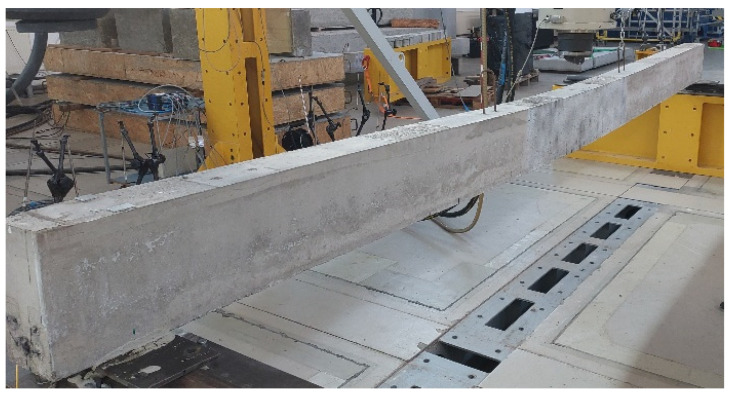
Typical failure mode of CFRP-prestressed beams.

**Figure 14 sensors-25-07357-f014:**
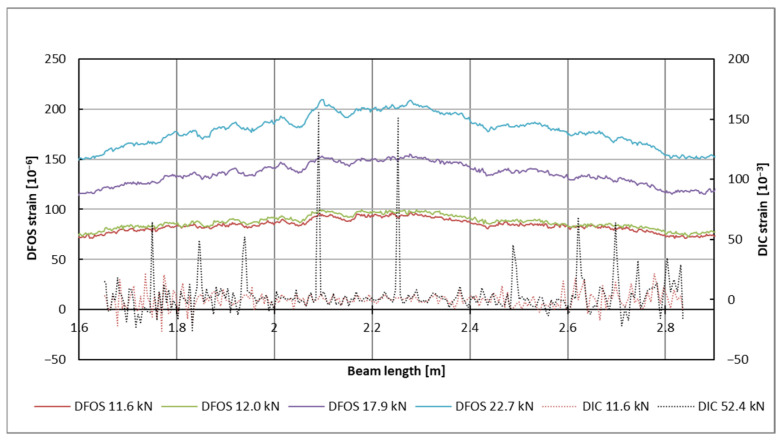
DFOS vs. DIC concrete strains comparison for the bottom zone of an exemplary beam.

**Figure 15 sensors-25-07357-f015:**
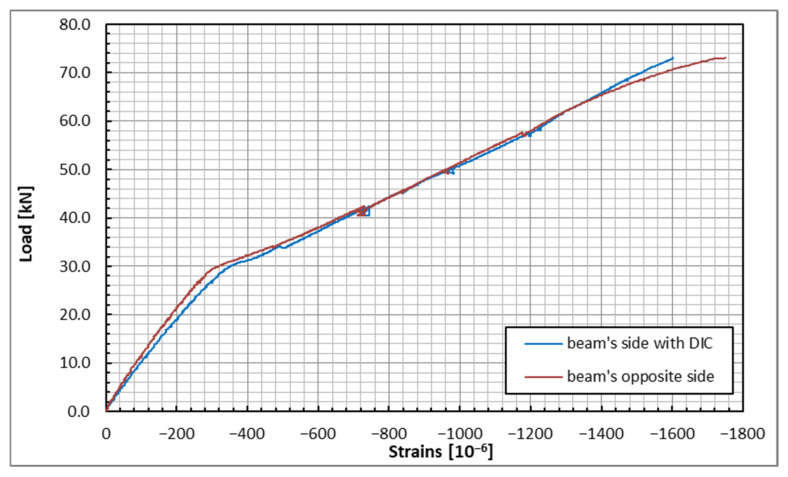
Load-strain plots in the top (**upper plot**) and bottom (**lower plot**) zones of the beam under the applied load steps (DFOS).

**Figure 16 sensors-25-07357-f016:**
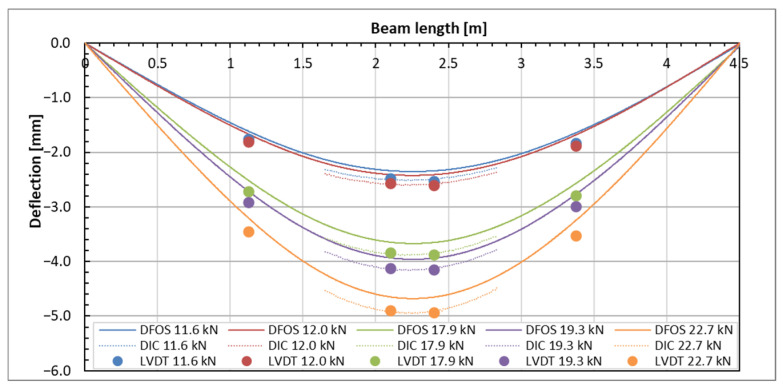
Deflections’ comparison of the exemplary beam determined by three methods for the load up to 22.7 kN.

**Figure 17 sensors-25-07357-f017:**
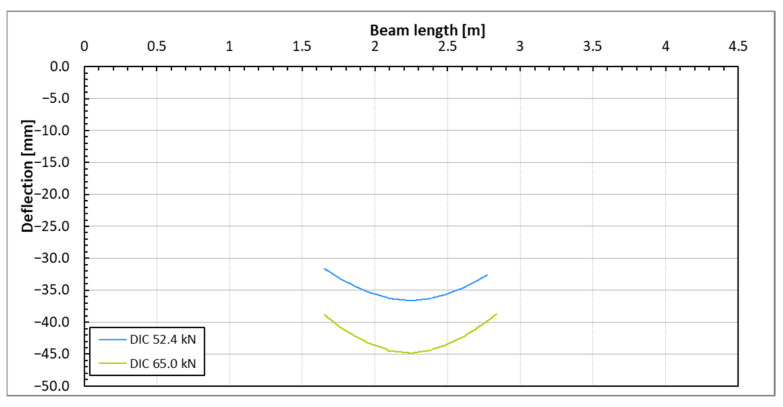
Supplementary beam deflection diagram in mid-span obtained by the DIC method for the highest loads.

**Figure 18 sensors-25-07357-f018:**
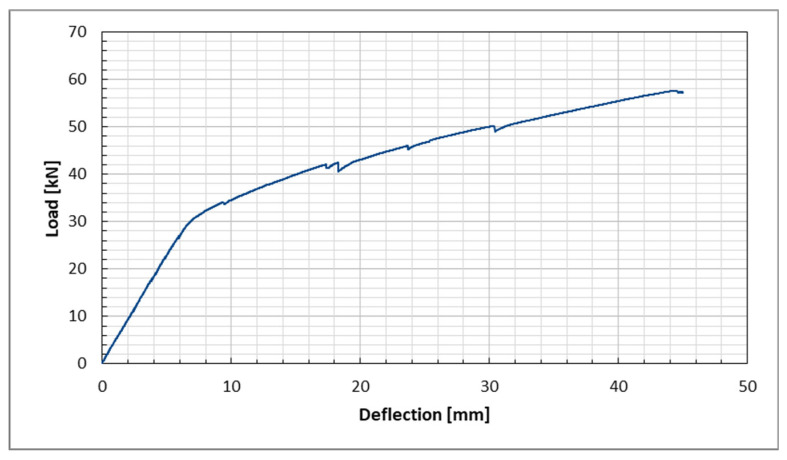
Load-deflection plot from DIC measurement.

**Figure 19 sensors-25-07357-f019:**
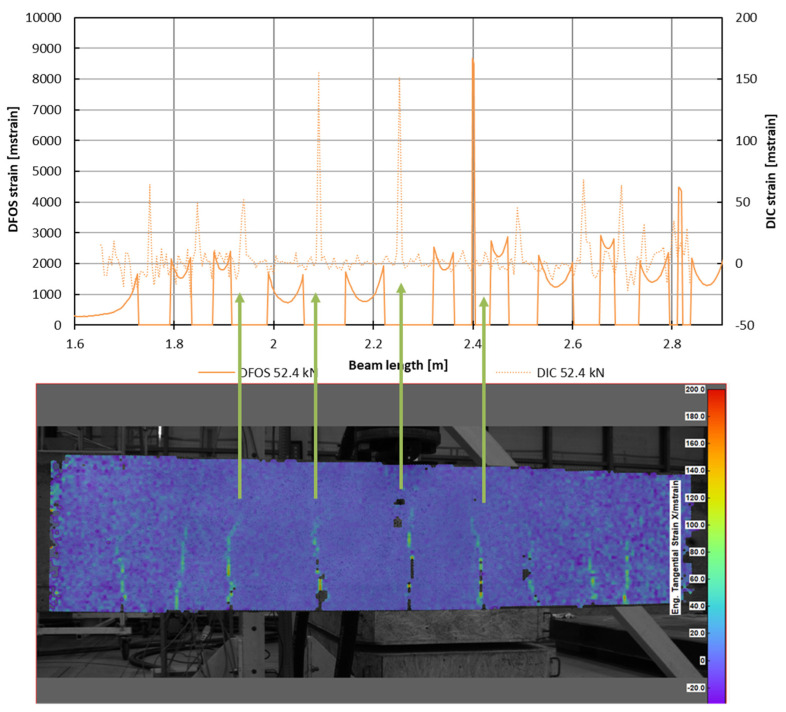
Comparison of the cracking picture of the selected beam based on DFOS (**top**) and DIC (**bottom**) measurements.

**Figure 20 sensors-25-07357-f020:**
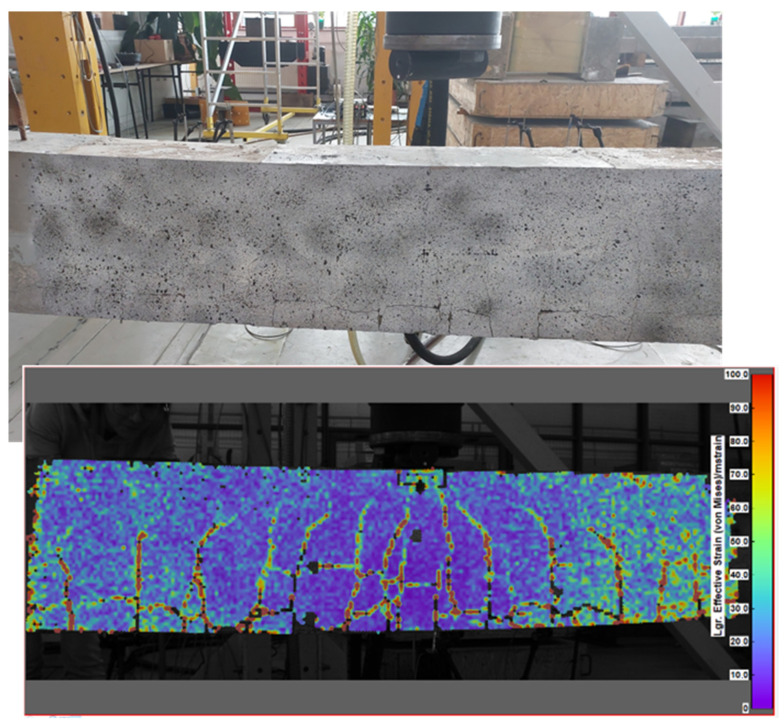
The real image of the failure mode taken conventionally (**top**) and the failure image (effective strain distribution map) obtained in DIC technology (**bottom**).

**Table 1 sensors-25-07357-t001:** Concrete mix design.

Component	Quantity [kg/m^3^]
CEM II/A-M (S-LL) 52.5 N cement	415
Sand 0/2	603
Grit 2/8	534
Grit 8/16	641
Superplasticizer	2.68
Admixture—set and hardening accelerator	0.83
Air-entraining admixture	0.6
Water	160

**Table 2 sensors-25-07357-t002:** Test loading levels and relevant design parameters.

Load Level	Test Load	Displacement Rate	Predicted Mid-Span Deflection	% of Cracking Moment	% of Design Capacity	Comments
	[kN]	[mm/min]	[mm]	[%]	[%]	
1	11.6	1.0	2.62	60.0	54.9	60% of the cracking moment
2	12.0	0.5	2.72	62.2	56.5	concrete decompression
3	17.9		4.05	92.7	80.6	$0.5\cdot \sqrt{{f’}_{c}}$ according to Table 3.2 [55]
4	19.3		5.10	100	86.3	cracking moment
5	22.7		9.01	115.6	100	design bending capacity
6	52.4		66.46	222.0	221.6	nominal bending capacity
7			failure			

**Table 3 sensors-25-07357-t003:** Selected parameters for DFOS measurement.

Parameter	Value	Unit
Distance range (standard mode)	up to 100	m
Spatial resolution (gauge spacing)	2.6	mm
Gauge length	2.6	mm
Strain measurement resolution	±1	10^−6^

**Table 4 sensors-25-07357-t004:** Displacement and strain maps for the three selected load levels.

Load	Tangential Strain (X)	Vertical Displacement (Y)
[kN]	[mstrain]	[mm]
22.7	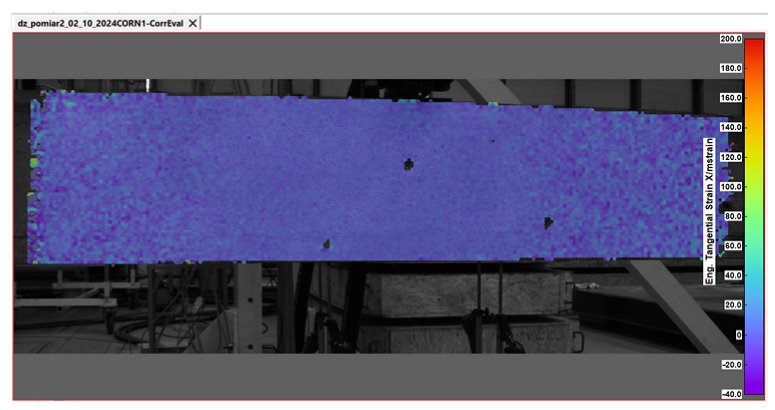	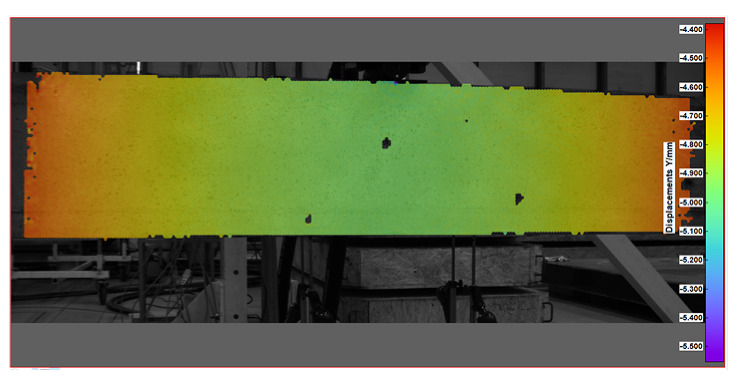
52.4	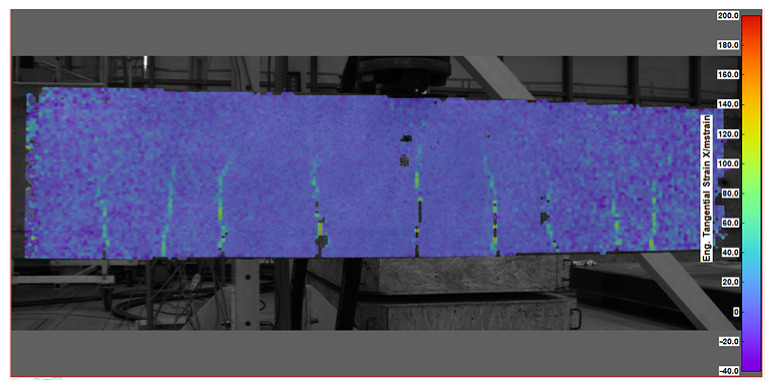	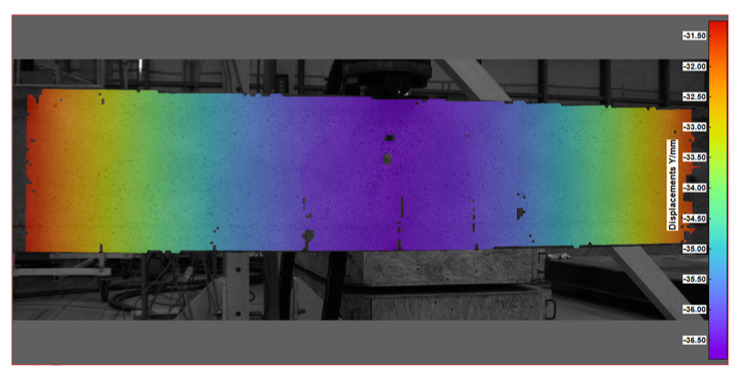
65.0	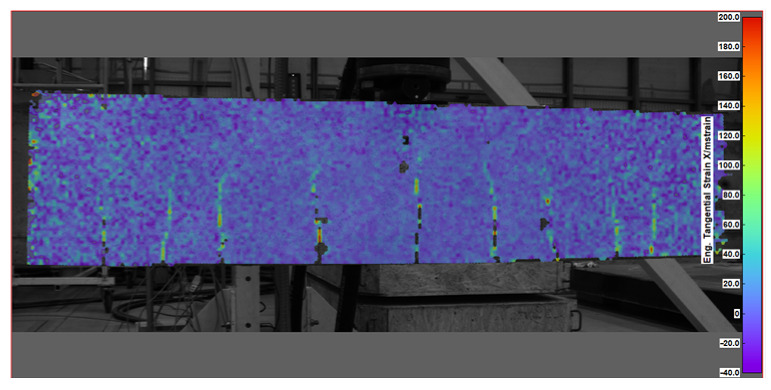	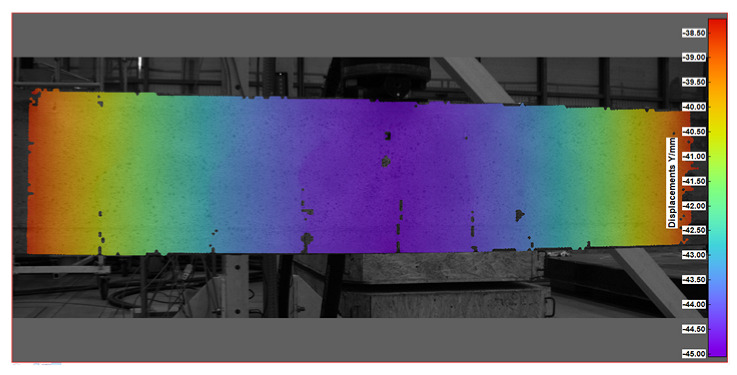

**Table 5 sensors-25-07357-t005:** Comparison of the mid-span deflections of the exemplary beam determined by three different methods.

Load Level	Load [kN]	Mid-Span Deflection [mm]				DFOS/LVDT	DIC/LVDT
		DFOS	DIC	LVDT	Piston		
1	11.6	1.97	2.17	2.11	3.66	93.4%	102.8%
2	12.0	2.05	2.19	2.19	3.72	93.6%	100.2%
3	17.9	3.07	3.22	3.22	5.01	95.3%	100.0%
4	19.3	3.34	3.50	3.50	5.34	95.4%	100.0%
5	22.7	3.92	4.10	4.10	6.00	95.6%	100.0%
6	52.4	-	32.73	32.73	36.67	-	100.0%
7	failure	-	64.16	-	64.16	-	-

## Data Availability

The raw data supporting the conclusions of this article will be made available by the authors on request.

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
