# Peer review of "Comparison of Distributed Fiber Optic Sensing and Digital Image Correlation Measurement Techniques for Evaluation of Flexural Behavior of CFRP-Prestressed Concrete Beams"

_sensors, 2025, doi:10.3390/s25237357_

Round 1
Reviewer 1 Report
Comments and Suggestions for Authors
This paper presents a comparative analysis of Distributed Fiber Optic Sensing (DFOS) and Digital Image Correlation (DIC) techniques for evaluating the flexural behavior of concrete beams prestressed with CFRP bars. The findings indicate that DFOS is more suitable for monitoring minor strains, whereas DIC demonstrates superior capability in measuring large strains and characterizing post-cracking behavior. The two methods prove to be complementary, enabling mutual validation and thereby enhancing the accuracy and comprehensiveness of structural health monitoring. The manuscript is logically structured, with a sound experimental design, well-supported data, and clear conclusions, demonstrating considerable academic merit and is recommended for acceptance.
Additionally, it is recommended to adhere to standard academic practice by defining the acronym "CFRP" in full upon its first occurrence in both the abstract and main text as "Carbon Fiber Reinforced Polymer (CFRP)".
Author Response
Comments 1:
Additionally, it is recommended to adhere to standard academic practice by defining the acronym "CFRP" in full upon its first occurrence in both the abstract and main text as "Carbon Fiber Reinforced Polymer (CFRP)".
Response 1:
In the abstract (line 21), an expansion of the abbreviation "CFRP" is provided. In the main body, the expansion of the acronym “CFRP” appears in line 104. The other abbreviations are introduced in the title, abstract, and main text.
Reviewer 2 Report
Comments and Suggestions for Authors
The novelty of this work requires further elaboration. The reviewer acknowledges that the experimental work in this paper is very thorough, with detailed reporting of the results. However, the research objective is unclear. Is it merely to compare two testing methods? Or to discuss the applicability of existing methods for the proposed new component type?
Neither of the two methods employed is original, and no customization of these methods was made. Furthermore, while the test results from both methods are presented and compared side-by-side, they are not sufficiently integrated to enable predictions of flexural performance (as the detection results are not quantitatively linked to the load-bearing capacity). Therefore, the authors' claim of a "combined application of DFOS and DIC" is not well supported.
The logic of the discussion and the conclusions in Section 6.1 need further refinement. The claimed measurement accuracy of DFOS and DIC at small strain levels should be validated against objective third-party data (e.g., local strain gauges), rather than relying solely on data stability. Additionally, the processing method of the raw DIC data can influence the representation of minor strains; thus, stating that DFOS is more accurate than DIC at small strains is not rigorous.
Finally, the methods discussed are primarily suited for laboratory testing, not for the monitoring and assessment of real-world structures. This limitation should be explicitly stated.
Author Response
Comments 1:
The novelty of this work requires further elaboration. The reviewer acknowledges that the experimental work in this paper is very thorough, with detailed reporting of the results. However, the research objective is unclear. Is it merely to compare two testing methods? Or to discuss the applicability of existing methods for the proposed new component type?
Response 1:
The authors agree with the Reviewer. The entire fourth and fifth paragraphs (last two) in the "Introduction" have been changed and amended as follows:
“This paper covers part of the research conducted by Rzeszów University of Technology (RUT) on the development and implementation of a novel method for prestressing precast concrete beams using carbon fibre-reinforced polymer (CFRP) bars. CFRP-prestressed concrete structures have lower deformation capacity than CFRP-reinforced concrete structures. To address this, high-strength concrete (HSC) was used to enhance the flexural performance of the prestressed beams. The research aimed to assess the flexural behaviour of precast prestressed concrete beams using a new CFRP prestressing technique. In this preliminary study, three high-strength concrete CFRP-prestressed beams were tested for flexural performance, with loading, strains, displacements, cracks, and failure modes recorded using three different measurement methods. The DFOS and DIC techniques monitored strains, cracks, and failure modes under load and verified the accuracy of the indirect deflection measurements obtained with these methods. Traditional linear variable differential transformers (LVDT) sensors were also utilised.
The scientific focus of this paper is combining DFOS and DIC to record strain and crack data in RC structures. According to the literature review (see p. 2), using both DFOS and DIC provides a more comprehensive understanding of how loaded structures behave. This, in turn, allows for more precise predictions of performance that traditional measurement systems cannot achieve. Furthermore, integrating both techniques enables result validation and the development of an effective monitoring system. The aim of this research is to assess the synergy of DFOS and DIC on a typical prefabricated RC beam, which features innovative materials and structural solutions (CFRP, HSC, prestressing) that are not relevant to this study. The main objectives are: to verify the measurement accuracy of both methods when used together; to explore the complementarity of results from each system; and to evaluate the applicability of each for assessing RC structure behaviour (cracks, strains, displacements). Since the combined use of DFOS and DIC is the primary focus, a detailed discussion on the behaviour of the new precast CFRP-prestressed beams under load until failure is provided elsewhere.”
Comments 2:
Neither of the two methods employed is original, and no customization of these methods was made. Furthermore, while the test results from both methods are presented and compared side-by-side, they are not sufficiently integrated to enable predictions of flexural performance (as the detection results are not quantitatively linked to the load-bearing capacity). Therefore, the authors' claim of a "combined application of DFOS and DIC" is not well supported.
Response 2:
Firstly, the combined use of DOFS and DIC, along with their synergy for assessing RC structures, has been demonstrated in a brief review (chapter 2 of the paper). Based on the experience described in Chapter 2 (which remains limited), the authors developed their own research programme to evaluate the accuracy and complementarity of both methods using their own research as an example. The main aim was to enhance the accuracy of detecting crack development in concrete elements reinforced with CFRP bars and to measure strains at individual load levels precisely. An additional objective was to verify indirect measurement of deflections using the DFOS method. Both goals were achieved, confirming the effectiveness of the combined measurement methods but also their limitations. Detailed conclusions from the use of the combined measurement system are presented in Chapter 7.
Comments 3:
The logic of the discussion and the conclusions in Section 6.1 need further refinement. The claimed measurement accuracy of DFOS and DIC at small strain levels should be validated against objective third-party data (e.g., local strain gauges), rather than relying solely on data stability. Additionally, the processing method of the raw DIC data can influence the representation of minor strains; thus, stating that DFOS is more accurate than DIC at small strains is not rigorous.
Response 3:
The authors fully agree with the Reviewer. Regrettably, this research programme did not include an additional (conventional) strain gauge measurement system. Accordingly, the authors recommend adding a refinement at the appropriate point in the text, in line with the Reviewer's comment. The suggested additional text is as follows:
“Summarising the strain measurement, it should be noted that the measurement accuracy of DFOS and DIC at low strain levels needs validation against objective third-party data (e.g., local strain gauges), rather than relying solely on data stability. Additionally, the processing method for raw DIC data can affect the representation of minor strains; therefore, claiming that DFOS is more accurate than DIC at low strains is not rigorous.”
Comments 4:
Finally, the methods discussed are primarily suited for laboratory testing, not for the monitoring and assessment of real-world structures. This limitation should be explicitly stated.
Response 4:
The Reviewer is generally correct; this may also stem from the review in Chapter 2. However, both methods have been applied to real bridge structures, as follows (for example):
- Howiacki, T., Sieńko, R., Bednarski, Ł., & Zuziak, K. (2023). Structural monitoring of concrete, steel, and composite bridges in Poland with distributed fibre optic sensors. Structure and Infrastructure Engineering, 20(7–8), 1213–1229. https://doi.org/10.1080/15732479.2023.2230558
- Tian, L., Zhao, J., Pan, B., & Wang, Z. (2021). Full‑Field Bridge Deflection Monitoring with Off‑Axis Digital Image Correlation. Sensors, 21(15), 5058. https://doi.org/10.3390/s21155058
- Halding, P. S., Schmidt, J. W., & Christensen, C. O. (2018). DIC‑monitoring of full‑scale concrete bridge using high‑resolution wide‑angle lens camera. In Proceedings of the 9th International Conference on Bridge Maintenance, Safety and Management (IABMAS) (pp. 1492–1499). CRC Press.
The authors also successfully carried out studies on a real bridge structure using both methods. The results of this research, demonstrating the feasibility of using both methods to monitor a real bridge, will be published soon.
Reviewer 3 Report
Comments and Suggestions for Authors
SUMMARY
This article compares measurement methods using distributed fiber-optic sensing and digital image correlation. The bending properties of prestressed reinforced concrete beams made of carbon fiber were evaluated.
The relevance of the study is due to the need to develop methods for evaluating the bending properties of prestressed reinforced concrete beams made of carbon fiber. Such constructions are relevant, in demand and effective. It should also be noted that it is necessary to collect new scientific data on their work using modern methods.
The work carried out by the authors is important and useful for modern science and for the practice of reinforced concrete products and structures. The results obtained are important for the development of construction science and for the improvement of the reinforced concrete industry.
The conducted article has a high level. It contains well-selected scientific literature for analysis, large-scale experimental studies have been carried out and important results have been obtained.
The reviewer believes that the article deserves support, but recommends correcting a few comments.
COMMENTS
1. The abbreviations should probably be removed from the brackets in the name. We are talking about the abbreviations "DFOS" and "DIC". This somewhat clutters up the title, and it already looks pretty heavy.
2. In the "Abstract" section, we would like the authors to begin not with a presentation of the methodology of scientific research, but to show a scientific problem. In other words, it should be reported that modern methods for measuring and evaluating the bending properties of precast reinforced concrete beams, especially those prestressed with carbon fiber reinforcement, are not effective enough and additional research is required to develop the theory and practice of this issue.
3. There are also no quantitative expressions of the result in the annotation. It is said that, for example, the DFOS method is more effective at detecting minor deformations, and the DIC method is better suited for measuring large deformations. However, I would like to understand the quantitative expression of these excesses. There should be numbers expressing these increases.
4. The "Introduction" section is missing some interesting research from the last 5 years. It would not be superfluous if the authors analyzed, for example, interesting works related to multi-density concretes used in reinforced concrete bendable beams. It is also interesting to consider in more detail the issue of differentiated fiber reinforcement with dispersed fiber. We would like the authors to add such reviews. It would not be superfluous to add 5-7 sources of literature on this topic and supplement the review with these works.
5. I would also like to see a more clearly formulated scientific problem at the end of the "Introduction" section, and from it the purpose and objectives of the research.
6. The methodological scheme of the study is missing. It would be good if the authors showed the parameters of the samples being manufactured, the factors being varied, and the indicators being determined. Such a flowchart would help structure the article.
7. In the discussion section, we would like to see an analytical summary table in which the results obtained will be compared with the results of other authors. The comparison criteria may include advantages, disadvantages, risks, and limitations of the data obtained.
8. I would like the authors to formulate in more detail the scientific and fundamental significance of the results obtained. What theoretical concepts about reinforced concrete structures have been developed? It would not be superfluous to add this.
9. The list of references, as already mentioned, should be supplemented with 5-7 sources over the past 5 years.

Author Response
Comments 1:
The abbreviations should probably be removed from the brackets in the name. We are talking about the abbreviations "DFOS" and "DIC". This somewhat clutters up the title, and it already looks pretty heavy.
Response 1:
The authors agree with the Reviewer. The abbreviations in the title in parentheses have been removed.
Comments 2:
In the "Abstract" section, we would like the authors to begin not with a presentation of the methodology of scientific research, but to show a scientific problem. In other words, it should be reported that modern methods for measuring and evaluating the bending properties of precast reinforced concrete beams, especially those prestressed with carbon fiber reinforcement, are not effective enough and additional research is required to develop the theory and practice of this issue.
Response 2:
The authors generally agree with the Reviewer, although there are various methods of writing abstracts. In our view, it is more important to present the paper's content accurately, as the scientific problem (mentioned by the Reviewer) is introduced in Chapter 1 (Introduction). This is particularly relevant because the paper's main scientific question is whether DFOS and DIC measurement techniques can be used together despite their respective limitations, as outlined in the summary.
Comments 3:
There are also no quantitative expressions of the result in the annotation. It is said that, for example, the DFOS method is more effective at detecting minor deformations, and the DIC method is better suited for measuring large deformations. However, I would like to understand the quantitative expression of these excesses. There should be numbers expressing these increases.
Response 3:
The effectiveness of both methods, along with the quantitative evaluation of measurement capabilities and accuracy, was detailed in separate subsections, presenting the results of DFOS and DIC measurements, respectively. However, these specific results are challenging to generalise quantitatively, as they pertain only to the particular case study described. Therefore, the conclusions (Chapter 7) offer a general comment rather than a detailed quantitative assessment. Nonetheless, the overall trend is correct, and we believe it can be presented in this manner.
Comments 4:
The "Introduction" section is missing some interesting research from the last 5 years. It would not be superfluous if the authors analyzed, for example, interesting works related to multi-density concretes used in reinforced concrete bendable beams. It is also interesting to consider in more detail the issue of differentiated fiber reinforcement with dispersed fiber. We would like the authors to add such reviews. It would not be superfluous to add 5-7 sources of literature on this topic and supplement the review with these works.
Response 4:
The paper compares and combines two novel measurement methods, aligning with the main theme of the Sensors journal. It demonstrates the implementation of both methods using a concrete beam reinforced with CFRP bars. However, the article does not focus on the concrete or its reinforcement; instead, it concentrates on the measurement methods. Consequently, the introduction mentions them in the context of studying concrete structures. Additionally, the authors found no existing research on using both measurement methods to examine concrete structures with multi-density concrete or those reinforced with dispersed fibres.
Comments 5:
I would also like to see a more clearly formulated scientific problem at the end of the "Introduction" section, and from it the purpose and objectives of the research.
Response 5:
The authors agree with the Reviewer. The entire fourth and fifth paragraphs (last two) in the "Introduction" have been changed and amended as follows:
“This paper covers part of the research conducted by Rzeszów University of Technology (RUT) on the development and implementation of a novel method for prestressing precast concrete beams using carbon fibre-reinforced polymer (CFRP) bars. CFRP-prestressed concrete structures have lower deformation capacity than CFRP-reinforced concrete structures. To address this, high-strength concrete (HSC) was used to enhance the flexural performance of the prestressed beams. The research aimed to assess the flexural behaviour of precast prestressed concrete beams using a new CFRP prestressing technique. In this preliminary study, three high-strength concrete CFRP-prestressed beams were tested for flexural performance, with loading, strains, displacements, cracks, and failure modes recorded using three different measurement methods. The DFOS and DIC techniques monitored strains, cracks, and failure modes under load and verified the accuracy of the indirect deflection measurements obtained with these methods. Traditional linear variable differential transformers (LVDT) sensors were also utilised.
The scientific focus of this paper is combining DFOS and DIC to record strain and crack data in RC structures. According to the literature review (see p. 2), using both DFOS and DIC provides a more comprehensive understanding of how loaded structures behave. This, in turn, allows for more precise predictions of performance that traditional measurement systems cannot achieve. Furthermore, integrating both techniques enables result validation and the development of an effective monitoring system. The aim of this research is to assess the synergy of DFOS and DIC on a typical prefabricated RC beam, which features innovative materials and structural solutions (CFRP, HSC, prestressing) that are not relevant to this study. The main objectives are: to verify the measurement accuracy of both methods when used together; to explore the complementarity of results from each system; and to evaluate the applicability of each for assessing RC structure behaviour (cracks, strains, displacements). Since the combined use of DFOS and DIC is the primary focus, a detailed discussion on the behaviour of the new precast CFRP-prestressed beams under load until failure is provided elsewhere.”
Comments 6:
The methodological scheme of the study is missing. It would be good if the authors showed the parameters of the samples being manufactured, the factors being varied, and the indicators being determined. Such a flowchart would help structure the article.
Response 6:
Since the main aim of the paper, as mentioned earlier (p. 5), is to compare the two measurement methods and evaluate the effectiveness of their combined use, the Reviewer's suggested flowchart is irrelevant. The article does not focus on the beams being studied (where variable parameters would be required), but rather on measurement methods, in accordance with the subject of the Sensors journal.
Comments 7:
In the discussion section, we would like to see an analytical summary table in which the results obtained will be compared with the results of other authors. The comparison criteria may include advantages, disadvantages, risks, and limitations of the data obtained.
Response 7:
The analytical summary table needs quantitative data on the results, which were not provided in the paper (see explanation on p. 3 above). The specific outcomes of our research are difficult to generalise quantitatively, as they relate only to the particular case study described. Therefore, the conclusions (Chapter 7) offer a general overview rather than a detailed quantitative assessment or comparison.
Comments 8:
I would like the authors to formulate in more detail the scientific and fundamental significance of the results obtained. What theoretical concepts about reinforced concrete structures have been developed? It would not be superfluous to add this.
Response 8:
The following text has been added at the end of the chapter 7 (while deleting the last bullet):
“The results showed that both DFOS and DIC measurement techniques can be used together to assess RC structures under flexural loading. However, their respective limitations should be considered. Nonetheless, the specific findings of this research are difficult to generalise quantitatively, as they relate only to the particular case study described. Therefore, the conclusions provide a broad comment rather than a detailed quantitative assessment. Nonetheless, the overall trend is accurate, and we believe it can be presented in this way. The authors also successfully conducted studies on a real bridge structure using both methods. The results of this research, demonstrating the feasibility of employing both approaches to monitor a real bridge, will be published soon.”
Comments 9:
The list of references, as already mentioned, should be supplemented with 5-7 sources over the past 5 years.
Response 9:
Please see comment 4.
Round 2
Reviewer 2 Report
Comments and Suggestions for Authors
The manuscript has been fully revised according to the reviewer's comments, and can be accepted for publication at the present form.
Reviewer 3 Report
Comments and Suggestions for Authors
The authors answered the reviewer's questions and made appropriate adjustments to the manuscript.
The reviewer has no further comments and the manuscript can be published in the journal in its current form.